# Ruminating, Eating, and Locomotion Behavior Registered by Innovative Technologies around Calving in Dairy Cows

**DOI:** 10.3390/ani13071257

**Published:** 2023-04-05

**Authors:** Ramūnas Antanaitis, Lina Anskienė, Giedrius Palubinskas, Karina Džermeikaitė, Dovilė Bačėninaitė, Lorenzo Viora, Arūnas Rutkauskas

**Affiliations:** 1Large Animal Clinic, Veterinary Academy, Lithuanian University of Health Sciences, Tilžės Str. 18, LT-47181 Kaunas, Lithuania; 2Department of Animal Breeding, Veterinary Academy, Lithuanian University of Health Sciences, Tilžės Str. 18, LT-47181 Kaunas, Lithuania; 3Health and Veterinary Medicine, University of Glasgow, Glasgow G12 8QQ, UK

**Keywords:** precision dairy farming, calving, innovation, dairy cattle

## Abstract

**Simple Summary:**

The use of technology to predict calving is increasingly being applied in animal reproduction. In our current study, we tested the hypothesis that there are correlations between facets of ruminating, eating, and locomotion behavior parameters registered by innovative technologies utilized by RumiWatch before and after calving. We found that rumination time, eating time, drinking time, and activity all decreased ten days before calving; drinking gulps decreased on the ninth and second days before calving; and down time decreased two days before calving, thus ensuring that the likelihood of calving is predicted from 10 days prior to the event.

**Abstract:**

The hypothesis for this study was that there are correlations between ruminating, eating, and locomotion behavior parameters registered by the RumiWatch sensors (RWS) before and after calving. The aim was to identify correlations between registered indicators, namely, rumination, eating, and locomotion behavior around the calving period. Some 54 multiparous cows were chosen from the entire herd without previous calving or other health problems. The RWS system recorded a variety of parameters such as rumination time, eating time, drinking time, drinking gulps, bolus, chews per minute, chews per bolus, activity up and down time, temp average, temp minimum, temp maximum, activity change, other chews, ruminate chews, and eating chews. The RWS sensors were placed on the cattle one month before expected calving based on service data and removed ten days after calving. Data were registered 10 days before and 10 days after calving. We found that using the RumiWatch system, rumination time was not the predictor of calving outlined in the literature; rather, drinking time, downtime, and rumen chews gave the most clearcut correlation with the calving period. We suggest that using RumiWatch to combine rumination time, eating time, drinking, activity, and down time characteristics from ten days before calving, it would be possible to construct a sensitive calving alarm; however, considerably more data are needed, not least from primiparous cows not examined here.

## 1. Introduction

Precision livestock farming (PLF) is a comprehensive term that encompasses technologies used with farm equipment (for example, sort gates, production measurements, or automated scales), as well as devices that can be worn by the animal (wearables). Although several solutions are called PLF, we define PLF for the purposes of this evaluation as a technique that automatically collects information in real time from each cow, also known as the per animal method [1]. Data from visual, sound, and movement sensors paired with algorithms can be used to monitor cow welfare, productivity, and management practices and can be integrated into welfare protocols [2]. PLF is acknowledged as essential for future dairy producers, since it allows for the constant monitoring of animal health and welfare throughout production, but suitable breakdowns and presentation of this plethora of data are essential so that farmers adopt and use the technologies [3].

Calving is a key time for both dairy cow and calf due to regrouping, nutrition changes, parturition, and the start of lactation [4]. Restless behavior, presumably associated with discomfort, is heightened in the final two hours before calving [5]. Dystocia and stillbirth have a negative impact on the dairy business. In reviews by Mee (2008) more than a decade ago, the dystocia rate in dairy cows was estimated at around 5% but was reported as higher in Holstein Friesians in the USA [6]. Indeed, in one study, this rate was as high as 28.6% in primiparous cows and 10.7% in multiparous cows [7]. Dystocia reduces milk supply and reproductive success, as well as increasing the likelihood of trauma and culling [8]. The number of dairy farms is reducing while the number of cows per farm is increasing; consequently, time for individual cow monitoring by stock workers is diminishing [9], yet ideally, close observation of cattle during the final gestation phase is required to detect the commencement of calving and thereby give assistance where indicated and prevent neonatal mortality [10]. Specialized delivery alarm systems, such as intravaginal devices expelled at the second stage of calving [11] and electronic data loggers attached to legs [4], have all been evaluated, but one study showed that rumination duration, feeding time, and dry matter intake were all reduced in the final 6 h before calving. Lying time decreased while lying bouts and activity increased [10].

The RumiWatch (RWS) combines a noseband sensor and a pedometer into a single system, making it a multipurpose system with a high level of usefulness, application, sensitivity, and specificity [12]. The RWS noseband sensor was successfully created and tested as a scientific monitoring device for the automated detection of rumination and eating behaviors in stable-fed dairy cows [13]. RWS has been evaluated in dairy herds all over the world both for confinement and grazing behaviors, and it has the potential to be utilized as a baseline for validating other animal behavior technologies [14,15]. Fadul et al. [16] in a relatively small study involving 20 heifers and 9 multiparous cows, sidered that the full RWS system can forecast the calving event to within 3 h. They warned that daily rumination summaries should be used with caution because one-hour classification summaries provide more detailed information. Calving alarms could be beneficial on-farm tools, but more research, perhaps concentrating on rumination and chewing, is needed to find reliable thresholds for decreased rumination time before calving [17].

The hypothesis for this study was that there are possible correlations between ruminating, eating, and locomotion behavior parameters registered by innovative technologies, particularly before calving as a means of predicting its timing, but also after calving.

## 2. Materials and Methods

### 2.1. Animals Farm and Feeding

This study was carried out at a Lithuanian dairy farm with 550 cows from 1 February 2022 to 11 November 2022 (location—55.911381565736, 21.881321760608195). The research was carried out in compliance with the Republic of Lithuania’s Animal Welfare and Protection Act (No. 108-2728; 2012, No. 122-6126). The study’s approval number was PK016965.

Some 60 cows were chosen from the entire herd. Cows were selected according to the following factors: cows have to be of the Lithuanian black and white breed, within 30 days of calving, with two or more lactations, 500–570 kg in body weight, and have an average productivity of previous lactation over 11,000 kg of milk per year, with dry cows consuming 14 kg of DM/day and fresh cows consuming 26 kg of DM/day. A general clinical examination revealed that none of the cows had clinical indications compatible with any disease. Cows with clinical symptoms of diseases (mastitis, metritis; *n* = 6) were excluded from the research. The total number of cows in this group was 54. The cows in the study had no calving problems or other health problems.

The cows were fed with total mix ration (TMR) twice a day, at 6:00 a.m. and 6:00 p.m., and were housed in a loose housing arrangement. The feed ration (Table 1) was balanced using the NorFor^®^ program (Agro Food Park 15, 8200 Aarhus N, Aarhus, Denmark) to meet the energy and nutritional requirements of a 500–570 kg Holstein cow (NRC 2001) (Table 2). Drinking water was freely available.

### 2.2. Measurements

#### 2.2.1. Instruments of Measurements

The RumiWatch sensors (RWS) consist of a liquid-filled pressure tube and a noseband halter with an integrated pressure detector. The pressure sensor in this system sends a pressure signal to the data recorder, which is mounted on the same halter and housed in a safe plastic box. There is also a sturdy memory card holder and an acceleration sensor for detecting triaxial head movements. At a frequency of 10 Hz, the acceleration values and pressure data are stored as binary files. The RumiWatch Manager program is linked to the halter through a wireless data transmitter, allowing for real-time data collection. The basic algorithms of the RWC software process the precise classification of behavioral 10 Hz pressure data features in a number of time summaries that can be selected. The algorithms recognize unambiguous pressure peak clusters produced by jaw motions, which are subsequently categorized based on their behavioral features [13].

RWS recorded rumination, eating, and locomotor behavior (rumination time, eating time, drinking time, drinking gulps, bolus, chews per minute, chews per bolus, activity up and down time, temp average, temp minimum, temp maximum, activity change, other chews, ruminate chews, and eating chews) (Table 3).

#### 2.2.2. Duration of Measurements

RWS sensors were placed one month before expected calving and removed ten days after calving. Data for statistical analysis were registered 10 days before and 10 days after calving.

### 2.3. Statistical Analysis

For the statistical analysis, we used version 25.0 of IBM SPSS 25.0 Statistics for Windows (IBM Corp., Armonk, NY, USA). Using descriptive statistics, normal distributions of variables were assessed using the Kolmogorov–Smirnov test. A linear regression equation was calculated to determine the statistical relationship between RumiWatch noseband sensor readings (dependent variables) by a day before and after calving (independent indicator), and means calculated in ANOVA accounting for the repeated effect of a cow. The Pearson correlation coefficient was calculated in order to define the linear relationship between the investigated variables. A linear regression equation was calculated to determine the statistical relationship between RumiWatch noseband sensor readings (dependent variable) and date (independent variable). If the probability was less than 0.05, it was thought to be reliable (*p* = 0.05).

## 3. Results

### 3.1. Differences in Rumination Time (RT) in the 10 Days before and after Calving

An analysis of our data revealed that the time of RT had a small tendency to increase just 2 days before calving and then from 3 to 10 days after calving. Significant mean differences in RT between the days before and after calving were detected between the calving day and all the days of the investigation. Due to a decline of rumination behavior on calving day 9, the rumination time differences ranged from 84.85 percent higher on the third day after calving, to 89.20 percent higher on the second day before calving, compared to the calving day, *p* < 0.001.

However, overall, there was relatively little change (y = −0.0537x + 21.453; R^2^ = 0.0341), i.e., the time of rumination decreased by only 0.0537 min/h in the period of ten days before and ten days after calving (*p* > 0.05) (Figure 1).

### 3.2. Differences in Eating Time (ET) in the 10 Days before and after Calving

An analysis of our data revealed that eating time (ET) had an overall tendency to decrease during the investigated period (by 0.0214 min/h), with a noticeable drop on the day of calving. Significant mean differences in eating time between the days before and after calving were detected only between the calving day and the fifth day after calving (41.35% higher at fifth day after calving, *p* < 0.05) (Figure 2).

### 3.3. Differences in Drinking Time (DT) in the 10 Days before and after Calving

An analysis of the drinking time data revealed that drinking time tended to increase quite steadily during the investigated period (y = 0.0156x + 0.3373; R^2^ = 0.4795). Significant mean differences were detected between the third day after calving compared with the tenth day before calving (64.52% higher on the third day after calving, and 61.29% higher on the third day after calving compared to the fourth day before calving (*p* < 0.05)) (Figure 3).

### 3.4. Differences in Drinking Gulp (DG) in the 10 Days before and after Calving

We estimated significant mean differences between the ninth day (55.28% higher) and the second day before calving (55.86% higher) compared to the calving day (*p* < 0.05). (Figure 4). An analysis of the drinking gulp data showed that drinking gulps tended to decrease steadily from 10 days before to 10 days after calving (y = −2,9571x + 172,31; R^2^ = 0.4179).

### 3.5. Differences in Chews per Bolus (CB) in the 10 Days before and after Calving

No significant mean differences were detected in chews per bolus during the investigated period, except a significant linear relation, where chews per bolus tended to increase (y = 0.1247x + 50,183; R^2^ = 0.5646) (Figure 5).

### 3.6. Differences in Activity in the 10 Days before and after Calving

An analysis of activity revealed a significant negative linear relation with days before and after calving, where activity had a tendency to decrease (y = −0.6337x + 77.244; R^2^ = 0.3027). Significant mean differences in activity ranged from 32.32% (higher on the first day after calving compared to the fifth day after calving, *p* < 0.01) to 24.10% (higher on the ninth day before calving compared to the sixth day after calving, *p* < 0.05) (Figure 6).

### 3.7. Differences in down Time in the 10 Days before and after Calving

Significant mean differences in down time ranged from 42.84% (higher on the calving day compared to one day before calving, *p* < 0.01) to 36.56% (lower on the seventh day after calving compared to one day before calving, *p* < 0.05) (Figure 7). An analysis of down time data showed that down time overall had a small tendency to increase (y = 0.1781x + 17.02; R^2^ = 0.1587) during the investigated period.

### 3.8. Differences in Other Chew in the 10 Days before and after Calving

Other chews showed an increase in the 2 days before calving, but this was only significant in the first day after calving. An analysis of other chew data showed no real overall change but a significant mean difference between the third day after calving (34.97% lower) and fifth day after calving (34.24% lower), compared to the first day after calving (*p* < 0.05) (Figure 8).

### 3.9. Differences in Ruminate Chew Bin the 10 Days before and after Calving

There were significant mean differences in ruminate chew, particularly in the period of 3 days to calving day and 3 days post-calving. These ranged from 39.37% (higher on the ninth day after calving compared to one day after calving, *p* < 0.001) to 26.96% (higher on the fifth day after calving compared to eight days before calving, *p* < 0.05) (Figure 9). An analysis of ruminate chew data showed that overall ruminate chew tended to increase (y = 28.794x + 1067.6; R^2^ = 0.5998) during the investigated period.

## 4. Discussion

Several animal welfare studies have been conducted in recent decades to investigate the behavioral changes that occur in cows before calving. As a result, maintenance behaviors such as locomotor and postural behavior (standing, lying down, and walking), as well as self-grooming and ingestive behavior (eating, drinking, and ruminating), have been studied [18]. The physiological unpredictability of the day of calving makes anticipating parturition difficult, raising the risk of unaided dystocia [18]. Further, the subsequent sequalae associated with dystocia reduce cow welfare and increase the likelihood of failed transitions [19].

In this study, we investigated how rumination, eating, and locomotion behaviors, as measured by the RumiWatch system, changed before and after calving. We found that time of RT had a small tendency to increase just 2 days before calving and then from 3 to 10 days after calving. These differences in rumination time suggest that it may be a predictor of calving time, but it is in the opposite direction to the literature. According to Soriani et al. and Clark et al., rumination time can be used to predict the day of calving [20]. Soriani et al. [20] found significant variations in RT over the transition phase (from 20 to +40 days). The changes in RT discovered by automated collars 8 h before calving suggested that continuous monitoring of rumination by mechanical devices is a beneficial tool in precision livestock farming for evaluating animal comfort and accurately predicting physiological or pathological problems [18]. In that study, rumination began to drop 10 h before calving [21]. Similarly, Clark et al. [22] reported a 33% decrease in RT duration between the day before calving and the day of calving. As a result, they suggested this behavior could be utilized to predict calving [23].

We found that while RT had a tendency to increase after calving (y = 0.5567x + 16.843; R^2^ = 0.6999), overall, it had a tendency to decrease by 0.0537 min/h ten days before and ten days after calving. According to the literature, rumination time is a sensitive indicator of dairy cow health that is used in automated systems to detect early disease onset [24,25]. Several factors influence rumination time, including enough physically effective neutral detergent fiber (peNDF) in the diet [24], forage inclusion and composition, and diurnal feed availability [25]. Furthermore, health problems, discomfort, and distress can all limit ruminating; in fact, a decrease in RT is regarded as a reliable indication of stress and sickness [24,25,26]. Calamari et al. (2014) linked slower increases in RT after calving to severe inflammation, implying the need of monitoring RT after calving to identify cows at increased risk of illness [25,27]. Since our findings using this RT parameter are somewhat different to others, we need some examination of eating and chewing times.

As has been reported in other studies, a considerable reduction in eating behavior was observed during the day of calving [27]. Indeed, this RumiWatch parameter, along with drinking time, downtime, and rumen chews, gave the most clearcut correlation with the calving period, but for all of these it showed the largest reduction on the day of calving. Significant (*p* < 0.05) mean differences in eating time between the days before and after calving were also detected, but only between the calving day and fifth day after calving (41.35% higher at fifth day after calving). Over this 20-day period, we found that eating time had a tendency to decrease by 0.0214 min/h. Intriguingly, rumination chews decreased consistently for the three days before calving before increasing to give an overall increase over the monitoring period. This decline in rumen chews has been reported by others [23]. Thus, while we saw no marked reduction in the RumiWatch parameter of RT, rumination chews and eating time were reduced around calving. The number of rumination chews is a component of rumination behavior and is linked to rumination time and DMI [28].

Significant (*p* < 0.05) mean differences in drinking time were detected between the third day after calving and the tenth day before calving (64.52% higher in on the third day after calving and 61.29% higher on the third day after calving), compared to the fourth day before calving. We estimated a significant (*p* < 0.05) mean of drinking gulp differences between the ninth day (55.28% higher) and the second day before calving (55.86% higher), compared to the calving day. Longer drinking durations around calving (24 min/d antepartum, 20 min/d postpartum) indicate that cows are thirstier due to weariness, stress, and water loss during calving and colostrum production [29]. Total drinking time increased in that study when cows transitioned from the pre- to post-calving period. This is also consistent with other literature, as large water losses occur as a result of increased postpartum milk production, particularly in early lactation [30]. As a result, we would have predicted a postpartum increase in drinking time with a maximum of minutes per day in early breastfeeding, rather than the reduction seen (8 min/d in early lactation) [17].

We also found significant (*p* < 0.05) mean differences in down time ranging from 42.84% (higher on the calving day compared to one day before calving, *p* < 0.01) to 36.56% (lower on the seventh day after calving compared to one day before calving). According to the literature, cows had a substantial increase in activity, indicating that they moved their heads more frequently in the last three hours before calving. During abdominal contractions, Jensen [13] described a frequent turn of the cows' heads towards the belly. Previous research found a general increase in activity in the last 24 h before calving [6,30], and our data appear to confirm this. Primiparous cows spend less time resting around calving than multiparous cows [31].

Significant mean differences in activity ranged from 32.32% (higher on the first day after calving compared to the fifth day after calving, *p* < 0.01) to 24.10% (higher on the ninth day before calving compared the sixth day after calving, *p* < 0.05), presumably due to the discomfort post-calving and perhaps also due to the stress of calf removal. Activity monitors can provide significant information on these important behaviors, which are linked to economic features such as health, milk output, and estrus detection. Activity-monitored metrics correlate with visual observations and provide correct information for dairy cattle management [31]. Beginning 8 h before calving, activity increased [21]. Miller et al. [32] and Borchers et al. [31] found an increase in activity and a decrease in rumination time in the 24 h preceding calving. During the transition period, lying and standing times varied depending on parity and response to postpartum health events [33]. Diseased cows had more recumbency and inactivity, which is consistent with their health status. In our previous investigation, there were differences in the peak of inactivity for healthy and ill cows, which were identical in response to cows fitted with a different pastern-mounted activity monitor [33].

These data imply that cows may grow more uncomfortable and spend less time lying down in the days preceding calving. This suggests an uncharacteristic change in normally occurring behaviors in the hours immediately preceding calving.

An analysis of other chew data showed significant mean differences only between the third day after calving (34.97% lower) and fifth day after calving (34.24% lower), compared to the first day after calving. Significant mean differences in chew ranged from 39.37% (higher on the ninth day after calving compared to one day after calving) to 26.96% (higher on the fifth day after calving compared to eight days before calving). Regarding parturition, lying episodes increase and rumination chews decrease [20]. The number of eating chews was generally consistent antepartum but decreased postpartum, possibly as a result of the growing amount of concentrates around calving and the occurrence of calving itself, which may cause discomfort for the cow [21].

## 5. Conclusions

Precision livestock farming systems, which are used to generate health and estrus alerts, effectively quantify behavioral changes around calving. Rumination time and feeding time show potential as techniques for identifying cows close to calving. These data are all defined by the RumiWatch system, which gives parameters that are clearly somewhat different to other parameters. Based on this, we recommend further research into physiological ranges with more clinically healthy cows. Moreover, for better prediction of calving by using sensor data and for using this dataset to make a prediction model using machine learning techniques, further research with a larger number of cows is needed. Future research on calving event prediction should focus on a longer period before calving and with more clinically healthy cows, using this dataset to make a prediction model using machine learning techniques. Moreover, calving events need to be tracked over shorter periods of time, so that farmers can receive notifications that can be used to make smart management decisions. Another use for calving prediction tools would be to distinguish between eutocial and dystocial calvings.

## Figures and Tables

**Figure 1 animals-13-01257-f001:**
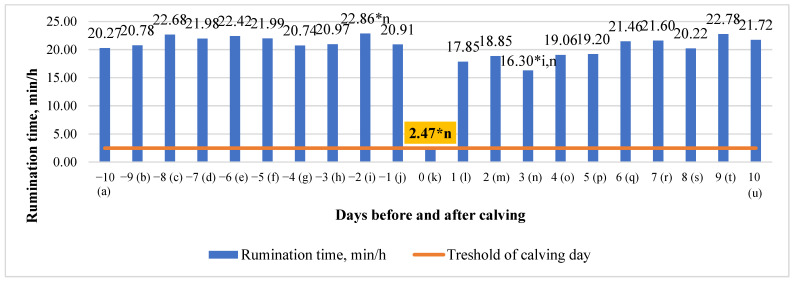
Differences in time of rumination between days before and after calving. The letters (a, b, c, d, e, f, g, h, i, j, k, l, m, n, o, p, q, r, s, t, u) indicate significant differences between days. * *p* < 0.05.

**Figure 2 animals-13-01257-f002:**
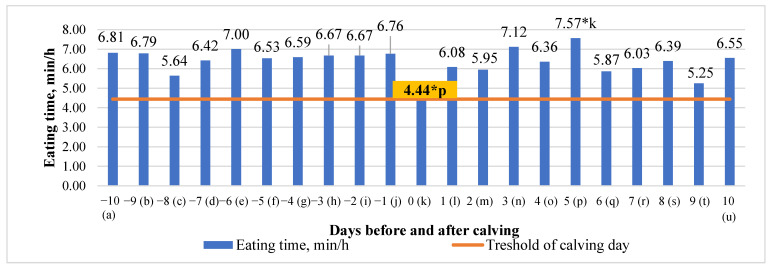
Differences in time of eating time (ET) before and after calving. The letters (a, b, c, d, e, f, g, h, i, j, k, l, m, n, o, p, q, r, s, t, u) indicate significant differences between days. * *p* < 0.05.

**Figure 3 animals-13-01257-f003:**
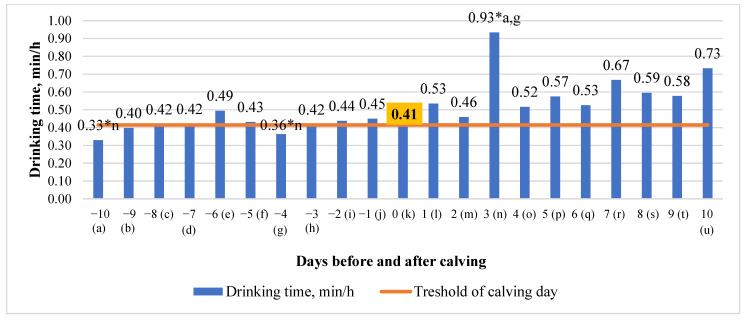
Differences in drinking time (DT) before and after calving. The letters (a, b, c, d, e, f, g, h, i, j, k, l, m, n, o, p, q, r, s, t, u) indicate significant differences between days. * *p* < 0.05.

**Figure 4 animals-13-01257-f004:**
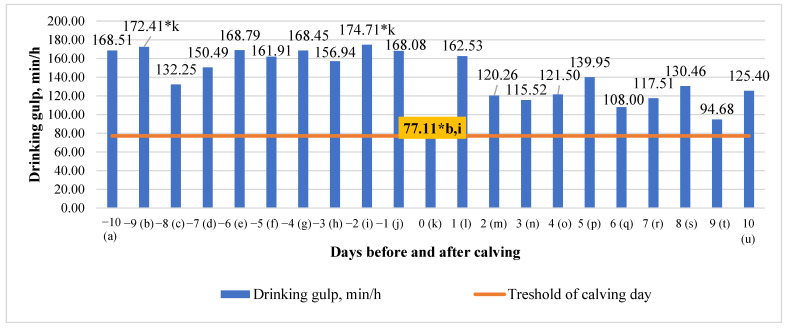
Differences in drinking gulp (DG) before and after calving. The letters (a, b, c, d, e, f, g, h, i, j, k, l, m, n, o, p, q, r, s, t, u) indicate significant differences between days. * *p* < 0.05.

**Figure 5 animals-13-01257-f005:**
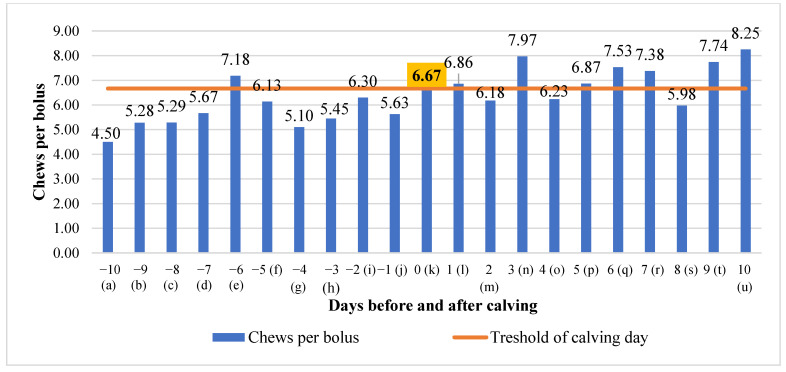
Differences in chews per bolus before and after calving. The letters (a, b, c, d, e, f, g, h, i, j, k, l, m, n, o, p, q, r, s, t, u) indicate significant differences between days.

**Figure 6 animals-13-01257-f006:**
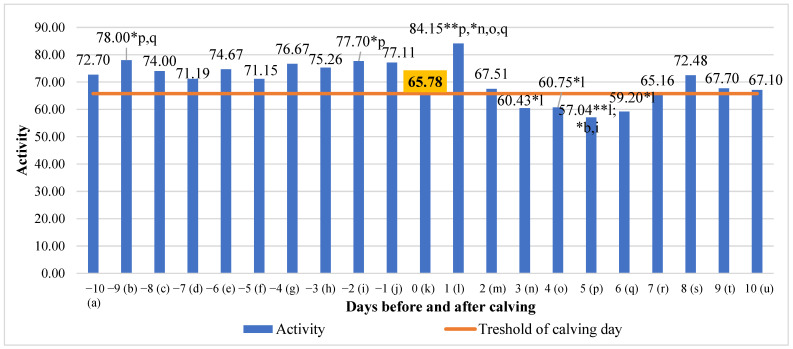
Differences in activity before and after calving. The letters (a, b, c, d, e, f, g, h, i, j, k, l, m, n, o, p, q, r, s, t, u) indicate significant differences between days. * *p* < 0.05, ** *p* < 0.01.

**Figure 7 animals-13-01257-f007:**
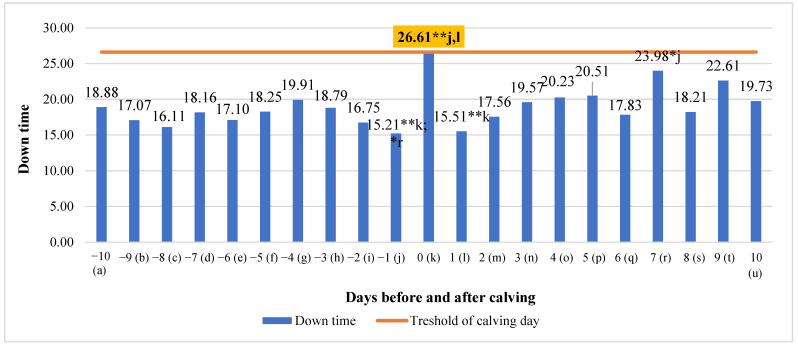
Differences in down time before and after calving. The letters (a, b, c, d, e, f, g, h, i, j, k, l, m, n, o, p, q, r, s, t, u) indicate significant differences between days. * *p* < 0.05, ** *p* < 0.01.

**Figure 8 animals-13-01257-f008:**
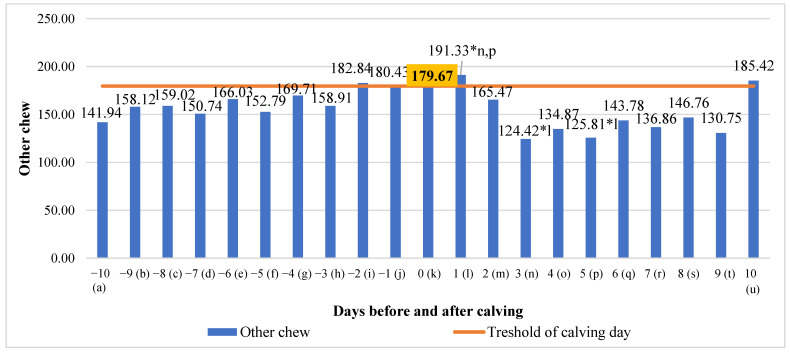
Differences in other chew before and after calving. The letters (a, b, c, d, e, f, g, h, i, j, k, l, m, n, o, p, q, r, s, t, u) indicate significant differences between days. * *p* < 0.05.

**Figure 9 animals-13-01257-f009:**
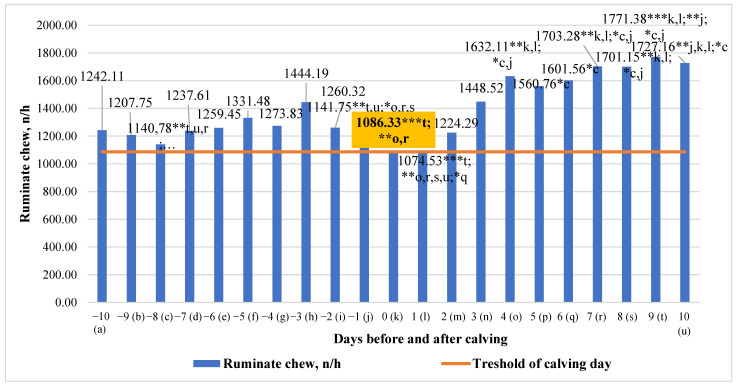
Differences in ruminate chew before and after calving. The letters (a, b, c, d, e, f, g, h, i, j, k, l, m, n, o, p, q, r, s, t, u) indicate significant differences between days. * *p* < 0.05, ** *p* < 0.01, *** *p* < 0.001.

**Table 1 animals-13-01257-t001:** The components of the total mix ration (TMR). Daily feeding for each animal.

Feed Component	Before Calving	After Calving
74% DM barley grain (kg)	0	3.5
56% DM corn grain (kg)	0	2.5
Rapeseed meal with 36% protein (kg)	1.2	2.5
Soy meal with 46% protein (kg)	0	1.5
Molasses from beets (kg)	0	0.5
27% DM grass silage (kg)	8	18
27% DM maize silage (kg)	1.2	23
Straws from wheat (kg)	7.5	0.5
BergaFat	0	0.200
Water (kg)	4.3	0.5
Mixture of grain (kg)		5.5
Mineral and vitamin supplement for lactating cows (kg)	0	0.250
Dry cow mineral and vitamin supplement (kg)	0.250	0

**Table 2 animals-13-01257-t002:** The chemical composition of cows before and after calving.

Parameter	Before Calving	After Calving
(%) dry matter	45.5	44.5
Dry matter consumption (DM) (kg DM/d)	12	28.2
Net energy for lactation (NEL) (MJ/kg DM)	5.4	7.1
Crude protein (g/kg DM)	102	175
Crude fat (g/kg DM)	30	50
Fatty acids (g/kg DM)	10	30
Rumen protein balance (g/kg DM)	11	25
Neutral detergent fiber (g/kg DM)	634	290
Starch (g/kg DM)	22	200
Acid detergent fiber (ADF) (g/kg DM)	175	183
Acid detergent lignin (ADL) (g/kg DM)	20	22
Sugar (g/kg DM)	30	60

**Table 3 animals-13-01257-t003:** Parameters recorded by RWS (Zehner et al. [13]).

Parameters	Description
Rumination time (RT)	Time spent on ruminating chews, including chewing breaks of up to 5 s
Eating time (ET)	Time spent chewing food, including breaks of up to 5 s
Drinking time (DT)	Time spent drinking, including up to 5 s pauses between gulps
Drinking gulps (DG)	Total amount of gulps taken while drinking
Chews per bolus (CB)	Chews performed during rumination between the regurgitation and swallowing of 1 bolus
Activity	Sum of the duration of all walking bouts presented as minutes within a given recording period
Down time	Time spent feeding with the head positioned downwards (min/h)
Rumination chews (RC)	Molars chewing during rumination for mechanical reduction in regurgitated materials into smaller masses
Other chews (OC)	Total number of trepidation bites and mastication chews made when eating

## Data Availability

The data presented in this study are available within the article.

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
