# Peer review of "Ruminating, Eating, and Locomotion Behavior Registered by Innovative Technologies around Calving in Dairy Cows"

_animals, 2023, doi:10.3390/ani13071257_

Round 1

Reviewer 1 Report (Previous Reviewer 3)

Dear authors, I think the manuscript has improved greatly. The English is much better and also the description of the study, the results and conclusions are now correct (no more suggesting that you did made a prediction model). I agree with publication in this form. 

One detail: in Line 342, 'hungrier' should be 'thirstier', I believe. 

Author Response

Dear Reviewer, 

Thank you very much for the comments and suggestions that helped us improve the manuscript.

We replaced 'hungrier'  with 'thirstier'

Best Regards, 

Prof. Ramunas Antanaitis 

Reviewer 2 Report (Previous Reviewer 1)

In my opinion, the manuscript has been significantly improved and now warrants publication in Animals.

Author Response

Dear Reviewer, 

Thank you very much for the comments and suggestions that helped us improve the manuscript.

Best Regards, 

Prof. Ramunas Antanaitis 

This manuscript is a resubmission of an earlier submission. The following is a list of the peer review reports and author responses from that submission.

Round 1

Reviewer 1 Report

This article (Animals-2182108) describes interesting results. The topic is actual, as the number of cows on farms increases, as the number of cows per stock worker. Zootechnicians cannot do without Precision Livestock Farming Systems.

The title and summary/abstract cover the content of the manuscript.

The description of materials and methods is complete. The work has been well designed and interpreted.

The interpretation of results is adequate and supported by the data.

In my opinion, this manuscript should be accepted for publication in Animals, after minor revisions.

I have some comments to the authors:

L85:  Benaissa et al., number 19, but in References, number 19 is Fadul et al… (L410)

L89: “…visualbservations.”  ??  Typing error, in my opinion.

Materials and Methods

Numbering of subsections is inaccurate (2.1, 2.2, 2.2.1, 2.4, 2.5), 2.3 is missing. This should be corrected.

L145: “… measurements -  .”  Typing error?

L155: “… (independent indicator) and  .”  Incomprehensible sentence.

Results

Regression equation on L166-167 does not agree with presented on Figure 1.

Numbering of figures is inaccurate:  Fig. 1, 2, 4, 5… (number 3 is omitted).

Author Response

Dear Reviewer, 

Authors are very thankful for the comments, which help us to improve the manuscript. All changes proposed have been included in the manuscript and highlighted in yellow and track changes.  

Best Regards, 

Prof. Ramunas Antanaitis 

Question  

Answers  

 L85:  Benaissa et al., number 19, but in References, number 19 is Fadul et al… (L410)

 Corrected

L89: “…visualbservations.”  ??  Typing error, in my opinion.

Corrected to – „visual observations”

Numbering of subsections is inaccurate (2.1, 2.2, 2.2.1, 2.4, 2.5), 2.3 is missing. This should be corrected.

Corrected

L145: “… measurements -  .”  Typing error?

Corrected

L155: “… (independent indicator) and  .”  Incomprehensible sentence.

Corrected

Regression equation on L166-167 does not agree with presented on Figure 1.

Corrected to. –„ ..increase after calving (y = -0,0537x + 21,453 R² = 0,0341),..”

Numbering of figures is inaccurate:  Fig. 1, 2, 4, 5… (number 3 is omitted).

Corrected

Reviewer 2 Report

Dear authors, I found the topic of your study to be very interesting, however, the quality of your submitted manuscript is very low, due to poor writing and unlikely/untrue statements.

Please find bellow my suggestions and remarks:

The simple summary is poorly written and misleading;

Lines 27-28: This sentence is confusing, please rephrase.

Lines 29-32: The same sentence is being repeated. An also ‘significant mean differences’ does not exist, it is ‘significant differences’ and we always insert the p value after stating this.

Abstract section: It lacks the main conclusion of the study. The way you present results is very hard to follow and read.

Please rewrite the entire abstract section, with aim, material and methods, results and discussion (presented clearer) and the main conclusions/implications of your study.

Introduction: As far as I know, PDT is PLF (precision livestock farming), please clarify and change throughout text;

Lines 106-123: Please rewrite this section, you have made this subsection in a haste, repeating information and having sentences that are not grammatically correct.

Lines 128-144: Please be more careful when describing the behavioural parameters, do consult with other authors who better defined they behavioural patterns, e.g. ‘Total number of gulps eaten while drinking’

Figure 3 is lacking;

Figure 6: showing the number of chews per bolus, I find this averages very unlikely, usually cows chew each bolus 20-30 times. Please use some published studies to validate/back-up your findings.

Discussions: Please be careful when stating ‘These investigations began with the goal of regulating parturient cows and preventing dystocia and metritis.’. How one regulates calving? And how predicting the time of calving throughout the use of PLFs is preventing dystocia and metritis in cattle?! Dear authors, with all the respect for your work, this sort of statements are unacceptable.

Rumination time it is not a biomarker as stated by the authors, please consult the available literature.

Line 313: Longer drinking durations around calving indicate that cows are hungrier? It makes absolutely no sense, are cows known to drink more water when they are hungry? The answer is no, water intake in the species it is corelated with dry matter intake, milk yield and outside temperature mainly.

I struggled the most with the discussion section to be honest, it is very poorly written and it makes little sense.

The conclusion section is poorly written, and main findings/implications of the study are lacking.

The conclusion is poorly written and makes absolutely no sense, and it sounds like ‘According to our hypothesis which was that changes in rumination, eating, and locomotion behavior before and after calving can be registered with innovative technologies, and the aim - to test this hypothesis, we planned to use innovative technologies to assess rumination, eating, and locomotion behavior 10 days before and 10 days after calving; we found from 9 to 3 days before calving reduction in rumination, eating behavior, such as rumination time, eating time, other chew, ruminate chew, drinking time and drinking gulp.’

In my opinion, the authors made this manuscript in a haste, without investing enough of their time to write a proper quality article, with clear findings and implications.

I strongly advise for this manuscript to be either completely rewritten in a more professional manner and resubmitted for revision, with authors investing time and efforts, or to be rejected.

Author Response

Dear Reviewer, 

Authors are very thankful for the comments, which help us to improve the manuscript. All changes proposed have been included in the manuscript and highlighted in yellow and track changes.  

Best Regards, 

Prof. Ramunas Antanaitis 

Question  

Answers  

The simple summary is poorly written and misleading;

We rewrote whole simple summary –

Simple summary. The use of calving detection technology is currently increasing in animal reproduction farming. In our current study, we decided to test the hypothesis: can we predict the onset of calving using sensor data registered by innovative technologies. According our results we concluded that using network machine-learning algorithms to combine rumination time, eating time, drinking, activity and down time characteristics resulted in sensitive and particular alarms at the daily from ten days before calving. We found hat sensor data collected by new technologies for ruminating, eating, and moving can help us predict calving time. For calving prediction, we can use rumination time, eating time, drinking, activity and down time. Rumination time, eating time, drinking time, and activity all decreased ten days before calving; drinking gulps decreased on the ninth and second days before calving; and down time decreased two days before calving. These data could be used by machine learning-based calving prediction algorithms for effective retrospective calving prediction”

Abstract section: It lacks the main conclusion of the study. The way you present results is very hard to follow and read.

Please rewrite the entire abstract section, with aim, material and methods, results and discussion (presented clearer) and the main conclusions/implications of your study.

We rewrote whole abstract section –

Abstract: Hypothesis for this study was that can we predict the onset of calving using sensor data registered by innovative technologies. Based on this the aim of this study was to identify indicators for calving prediction using sensor data, described rumination, eating, and locomotion behavior around calving as registered by innovative technologies.

54 cows were chosen from the entire herd without calving problems or other health problems. For registration of rumination, eating, and locomotion behaviors, the Rumiatch sensors (RWS) was used. RWS recorded rumination, eating, and locomotor behavior (rumination time, eating time, drinking time, drinking gulps, bolus, chews per minute, chews per bolus, activity up and down time, temp average, temp minimum, temp maximum, activity change, other chews, ruminate chews, eating chews). RWS sensors were placed one month before expected calving and removed ten days after calving. Data were registered 10 days before and 10 days after calving.

We found that rumination time had a tendency to decrease by 0.0537 min/h ten days before and ten days after calving. On calving day, we recorded a sudden drop, the eating time had a tendency to decrease by 0.0214 min/h ten days before and ten days after calving. Differences in drinking time were found between the third day after calving and the tenth day before calving, but in general drinking time was lower 10 days before calving day, compared to 10 days after calving.  Drinking gulps differed between the ninth and second days before calving compared to the calving day. Activity was found to be significant between the first, third and fith day after calving and the ninth and second day before calving. Atctivity had a tendency to decrease by 0.6337 10 days before and after calving. Down time between had a tendency to decease aprroximatlety 2 days before calving.

Based on the results of this study, we can say that sensor data collected by new technologies for ruminating, eating, and moving can help us predict calving time”

Introduction: As far as I know, PDT is PLF (precision livestock farming), please clarify and change throughout text;

Corrected to - PLF (precision livestock farming)

Lines 106-123: Please rewrite this section, you have made this subsection in a haste, repeating information and having sentences that are not grammatically correct.

Corrected to – „60 cows were chosen from the entire herd. Cows were selected according to the following factors: cows has to be within 30 days of calving of the Lithuanian black and white breed, with two or more lactations, 500 – 570 kg body weight, and an average productivity of previous lactation - 11000 kg milk per year, with dry cows consuming 14 kg DM/day and fresh cows consuming 26 kg DM/day. A general clinical examination revealed that none of the cows had clinical indications compatible with any disease. Cows having clinical symptoms of diseases (mastitis, metritis) (n = 6) were excluded from the research. The total number of cows in this group was 54. The cows in the study had no calving problems or other health problems”

Lines 128-144: Please be more careful when describing the behavioural parameters, do consult with other authors who better defined they behavioural patterns, e.g. ‘Total number of gulps eaten while drinking’

Corrected according Zehner et al.

Figure 3 is lacking;

Corrected

Figure 6: showing the number of chews per bolus, I find this averages very unlikely, usually cows chew each bolus 20-30 times. Please use some published studies to validate/back-up your findings.

We added information in methodology section - Table 3.Parameters recorded by RWS (Zehner et al. [17]).   

“Chews per bolus (CB) - Chews performed during rumination between the

regurgitation and swallowing of 1 bolus”

Discussions: Please be careful when stating ‘These investigations began with the goal of regulating parturient cows and preventing dystocia and metritis.’. How one regulates calving? And how predicting the time of calving throughout the use of PLFs is preventing dystocia and metritis in cattle?! Dear authors, with all the respect for your work, this sort of statements are unacceptable.

We corrected and deleted this sentence

Rumination time it is not a biomarker as stated by the authors, please consult the available literature.

We corrected to – “Rumination time is a sensitive indicator of dairy cow health that is used in automated systems to detect early disease onset”

Line 313: Longer drinking durations around calving indicate that cows are hungrier? It makes absolutely no sense, are cows known to drink more water when they are hungry? The answer is no, water intake in the species it is corelated with dry matter intake, milk yield and outside temperature mainly.

We agree with you and deleted this sentence

I struggled the most with the discussion section to be honest, it is very poorly written and it makes little sense.

We corrected all discussion section with more concentration in our explanation of our results.

The conclusion is poorly written and makes absolutely no sense, and it sounds like ‘According to our hypothesis which was that changes in rumination, eating, and locomotion behavior before and after calving can be registered with innovative technologies, and the aim - to test this hypothesis, we planned to use innovative technologies to assess rumination, eating, and locomotion behavior 10 days before and 10 days after calving; we found from 9 to 3 days before calving reduction in rumination, eating behavior, such as rumination time, eating time, other chew, ruminate chew, drinking time and drinking gulp.

We rewrote the entire conclusion section – 

“Precision livestock farming systems, which are used to generate health and estrus alerts, effectively quantify behavioral changes around calving. Based on the results of this study, we can say that sensor data collected by new technologies for ruminating, eating, and moving can help us predict calving time. For calving prediction, we can use rumination time, eating time, drinking, activity and down time. Rumination time, eating time, drinking time, and activity all decreased ten days before calving; drinking gulps decreased on the ninth and second days before calving; and down time decreased two days before calving. These data could be used by machine learning-based calving prediction algorithms for effective retrospective calving prediction.

Using network machine-learning algorithms to combine rumination time, eating time, drinking, activity and down time characteristics resulted in sensitive and particular alarms at the daily from ten days before calving. Future research on calving event prediction should focus on longer period before calving and with more clinically healthy cows, using this dataset to make a prediction model using machine learning techniques. Also, calving events need to be tracked over shorter periods of time that farmers can get notifications they can use to make smart management decisions. Another use for calving prediction tools would be to distinguish between eutocial and dystocial calvings”

In my opinion, the authors made this manuscript in a haste, without investing enough of their time to write a proper quality article, with clear findings and implications.

Strongly advise for this manuscript to be either completely rewritten in a more professional manner and resubmitted for revision, with authors investing time and efforts, or to be rejected.

We corrected all parts of manuscript –

We changed title of manuscript – “Can We Predict Calving By Using Innovative Technologies Registered For Ruminating, Eating, And Locomotion Behavior?”

Hypothesis and aim – “Hypothesis for this study was that can wepredict the onset of calving using sensor data registered byinnovative technologies. Based on the hypothesis, the aim of this study was to identify indicators for calving prediction using sensor data, described rumination, eating, and locomotion behavior around calving as registered by innovative technologies”

Also, according that we made correction in whole manuscript.

Reviewer 3 Report

I am sorry to say that I find this article not ready for publication. 

The hypothesis and study is not new: we already know that behaviour changes before calving, and that we can measure this with sensors. This has been shown in several studies. An interesting hypothesis would be whether we can predict the onset of calving with sensor data.

The data that is gathered is interesting, and probably enough to answer the question above. But, the authors have only used descriptive analyses to show the behaviours before and after calving. Differences between days are calculated but those are not very interesting, if you do not use these results to make a prediction. With the use of machine learning techniques, you could develop a prediction model for the onset of calving. That would be interesting, and worth a publication, if you get nice results with these data - otherwise you need to gather more data of calving cows wearing sensors.  The authors even refer to studies where predictions have been done, e.g. Miller et al. 2020, Titler et al., 2015, and Ouellet et al., 2016.

I would advise the authors to change their hypothesis and the title to 'can we predict the onset of calving using sensor data?' and to use this dataset to make a prediction model using machine learning techniques. 

Author Response

Dear Reviewer, 

Authors are very thankful for the comments, which help us to improve the manuscript. All changes proposed have been included in the manuscript and highlighted in yellow and track changes.  

Best Regards, 

Prof. Ramunas Antanaitis 

Question  

Answers  

The hypothesis and study is not new: we already know that behaviour changes before calving, and that we can measure this with sensors. This has been shown in several studies. An interesting hypothesis would be whether we can predict the onset of calving with sensor data.

We corrected the title, hyphotesis and aim of this study. Also, we rewrote the discussion and conclusion sections.

The data that is gathered is interesting, and probably enough to answer the question above. But, the authors have only used descriptive analyses to show the behaviours before and after calving. Differences between days are calculated but those are not very interesting, if you do not use these results to make a prediction. With the use of machine learning techniques, you could develop a prediction model for the onset of calving. That would be interesting, and worth a publication, if you get nice results with these data - otherwise you need to gather more data of calving cows wearing sensors.  The authors even refer to studies where predictions have been done, e.g. Miller et al. 2020, Titler et al., 2015, and Ouellet et al., 2016.

Prediction equations or models are based on correlations. In our research work correlations between  trait from RumiWatch noseband sensor readings and a day before and after calving were  low and not statistically significant. As You can see in the figure bellow.

Regression analysis was used  for determination of the relationship between a single dependent ( trait from RumiWatch noseband sensor readings) variable and one  independent /predictor (a day before and after calving) variables. The analysis yields a predicted value for the criterion resulting from a linear combination of the predictors.

We think for such a modelling analysis a higher number of cows should be investigated.

I would advise the authors to change their hypothesis and the title to 'can we predict the onset of calving using sensor data?' and to use this dataset to make a prediction model using machine learning techniques. 

We changed title to – “Can We Predict Calving By Using Innovative Technologies Registered For Ruminating, Eating, And Locomotion Behavior?”

Hypothesis and aim to – “Hypothesis for this study was that can we predict the onset of calving using sensor dataregistered by innovative technologies. Based on the hypothesis, the aim of this study was to identify indicators for calving prediction using sensor data, described rumination, eating, and locomotion behavior around calving as registered by innovative technologies”

Round 2

Reviewer 2 Report

Dear authors, thank you for taking into account my suggestions. I support the publication of your manuscript.

Author Response

Dear Reviewer,

Thank you very much for your positive decision.

Best Regards,

Prof. dr. Ramunas Antanaitis 

Reviewer 3 Report

I am sorry, but I still have the same objections against publishing your data in this stage. The title and hypothesis was changed, but the analysis is the same. You are not making predictions. Identifying factors that can be used in a prediction model, is the first step in that process, and you did only that; to me, that is not enough. If you cannot make a prediction model because you do not have enhough data, than you have to repat he study and gather more data. If you make a prediction model, you might be able to publish it. But only descriptive analysis of factors that change around calving has not much news value and in my opinion is not enough to publish. 

Author Response

Dear Reviewer,

We corrected the title, hypothesis, aim, and discussion, and changed the word "prediction" to "correlation."

Best Regards,

Prof. dr. Ramunas Antanaitis